# Resource diversity and supply drive colonization resistance

Ethan S. Rappaport[1,2], Renato Mirollo[1], Babak Momeni[2]*

1 Mathematics Department, Boston College, Chestnut Hill, Massachusetts, United States of America,
2 Biology Department, Boston College, Chestnut Hill, Massachusetts, United States of America

* momeni@bc.edu

## Abstract

The human microbiota plays a key role in resisting the colonization of pathogenic microbes, a process known as colonization resistance. However, there is a need to better understand the mechanisms by which colonization of invaders is blocked. Environmental resource supply and resource diversity are essential factors in forming these communities but testing how the environment affects resistance in natural communities is challenging. Here we use a consumer-resource model and computational invasion simulations to investigate how environmental resource diversity and supply affect the richness-resistance relationship, overall colonization resistance, and cross-feeding dynamics. We find a non-monotonic trend between species richness and resistance, shaped by environmental characteristics. Our results show that colonization resistance is negatively correlated with both resource supply and resource diversity except when resource supply is limited. Lastly, we observe that cross-feeding weakens colonization resistance by increasing the diversity of available resources, but this trend disappears with limited resource supply. This work provides insights about colonization resistance in microbial communities of consumers, resources, and resource conversion and exchange.

## Author summary

Resident microbial communities, such as those inhabiting different parts of our body, can resist colonization by other microbes, a property called colonization resistance. Colonization resistance is important for protecting us from pathogens, bringing up the need to better understand the mechanisms that affect it. Here we use a consumer-resource model to investigate how resources supplied in the environment can influence invasion outcomes including colonization resistance. We implement a computational invasion assay and simulate many instances of resident communities encountering invaders to infer how different parameters such as the supply of resources or exchange of metabolites between species

**Data availability statement:** The code used to generate the data and figures in this study are available in a public repository: https://github.com/esrapp92202/envfact.

**Funding:** ESR received an Undergraduate Research Fellowship from Boston College. BM received funding from the National Science Foundation (NSF MCB, Grant No. 2430384). BM and RM received funding from the Schiller Institute for Integrated Science and Society at Boston College (SI-GECS grant). The funders had no role in study design, data collection and analysis, decision to publish, or preparation of the manuscript.

**Competing interests:** The authors have declared that no competing interests exist.

affect colonization resistance. We find that colonization resistance is negatively correlated with both resource supply and resource diversity, except when resource supply is limited. We also show that cross-feeding between species weakens colonization resistance by increasing the diversity of available resources, but this trend disappears with limited resource supply. Collectively, our work highlights the impact of resources in shaping colonization resistance, offering useful insights that can guide future efforts to control colonization resistance.

## Introduction

Whether an invader can successfully colonize an ecosystem depends both on the resources available in the environment as well as direct and indirect interactions with the resident community already present in that ecosystem [1]. The combined ability of an environment and the resident community to prevent the establishment of an invader is known as colonization resistance. For example, the pathogen *Clostridioides difficile* generally fails to colonize in a healthy human intestinal microbiota but succeeds in hosts recently treated with antibiotics [2–4]. Likewise, colonization of the opportunistic pathogen *Staphylococcus aureus* is inhibited by the nasal microbiota resident *Staphylococcus epidermidis* [5,6].

Colonization resistance plays a key role in protecting hosts from pathogenic invasion. The ability to manipulate the microbiome's composition can also benefit human health by treating conditions associated with dysbiosis of our resident microbiota [7,8], such as Crohn's disease [9], obesity [10,11], and depression [12]. However, the mechanisms of colonization resistance are not well understood, and assessing the impact of interventions *in vivo* is challenging [13]. Various model systems have been proposed to study colonization resistance, yet each faces trade-offs between control and realism [14].

Deciphering emergent community functions such as colonization resistance requires a fundamental understanding of species-level interactions. Microbial species in a community interact directly and indirectly through strategies such as resource competition, antimicrobial production, and contact-dependent inhibition [15,16]. Focusing on the abiotic factors that shape a community—such as environmentally supplied nutrients—may present key insights into emergent functions such as colonization resistance [17]. For example, environments with fewer supplied resources are expected to support a smaller niche space, making successful invasion more difficult. The extent to which resource diversity affects species richness and community resistance, or how cross-feeding modulates colonization resistance, remains unclear [18,19].

Previous analyses of the relationship between species richness and colonization resistance have led to various and often contradictory results, where both positive and negative correlations appear across ecosystems [20,21]. Another study found non-monotonic richness-resistance relationships, with a positive correlation for low species richness and a negative correlation for high richness [22]. The variety of

richness-resistance relationships has been termed the *invasion paradox*, describing the several concurrently supported theories for both a positive and a negative correlation between resident biodiversity and invasion success [23]. Given the importance of microbial diversity in community functionality, understanding which factors determine the shape of the richness-resistance curve is imperative to designing artificial communities and analyzing the composition of our microbiota [24–28].

Due to the difficulty in characterizing and controlling complex microbial communities, the general principles governing these ecosystems are poorly understood. Thus, mechanistic studies in microbial community ecology are typically either general and based on theoretical models or specific to a narrow set of species and conditions [29–31]. We take the former approach in this work, highlighting general insights using *in silico* models. The Lotka-Volterra model, a simple model of direct species interactions, has been used in the past to model community dynamics; however, such a model is incapable of accounting for resource preference and cross-feeding [32]. Instead, we employ a modified version of MacArthur's consumer-resource model, which captures the dynamics of species populations and resource concentrations over time [33]. We directly study how environmental resource influx and cross-feeding shape invasion dynamics within a consumer-resource model. Similar analyses have been performed using other models to decipher how direct interactions affect community colonization resistance, finding that more facilitative resident communities are more resistant to invasion [34,35]. Even though Lotka-Volterra dynamics can emulate low-nutrient environments, they fail to consider higher-order indirect interactions, such as resource competition, consumption trade-offs, and metabolic cross-feeding [36]. To account for higher-level interactions, we use a consumer-resource model based on these abiotic factors [37].

We begin our investigation by testing whether the richness-resistance relationship depends on the amount and diversity of resources supplied. After invading communities of varying richness, we find a non-monotonic trend between species richness and resistance shaped by environmental characteristics. We next ask whether the environment impacts colonization resistance trends. To elucidate how the environment affects the richness-resistance relationship, we systematically vary *resource supply*, or the total input rate of supplied resources, and *resource diversity*, the number of resource types over which this supply is distributed. We observe that these two factors have an independent and interdependent effect on the richness-resistance relationship, offering valuable insights into the environmental conditions that predispose communities to positive, negative, or non-monotonic richness-resistance curves. We find that resistance is negatively correlated with resource supply and is typically—but not universally—negatively correlated with resource diversity. However, when resource supply is limited, we observe a surprising positive correlation between resource diversity and resistance. Finally, we test cross-feeding parameters which control the amount and diversity of metabolic byproducts. We observe that high resource diversification via cross-feeding weakens colonization resistance, but this trend disappears with limited resource supply. These results contribute towards a greater understanding of how environmentally supplied resources and cross-feeding dynamics help to determine the colonization resistance of a community.

## Methods

### Microbial consumer-resource model

We use the following consumer-resource model, visualized in Fig 1a-1b and parametrized as listed in Table 1. Each species is parameterized by a weighted set of resource preferences and a minimum resource uptake threshold to maintain its population size. Species population densities $N_i$ grow or shrink according to the alignment between their preferences and the resources present, with a minimum uptake $m_i$ to achieve zero growth. Resource concentrations $R_\alpha$ are determined by an environmental influx rate $K_\alpha$, an environmental dissipation rate $\gamma$, and species consumption $c_{i\alpha}$. Additionally, we incorporate cross-feeding dynamics using byproduct leakage proportion $l$ and a metabolic conversion matrix $D$ as described in [38].

$$\frac{dN_i}{dt} = N_i \left[ (1-l) \sum_\alpha c_{i\alpha} R_\alpha - m_i \right]$$
$$\frac{dR_\alpha}{dt} = K_\alpha - R_\alpha \left( \gamma + \sum_i c_{i\alpha} N_i \right) + l \sum_\beta \left( R_\beta D_{\alpha\beta} \sum_i c_{i\beta} N_i \right)$$

(1)

PLOS Computational Biology

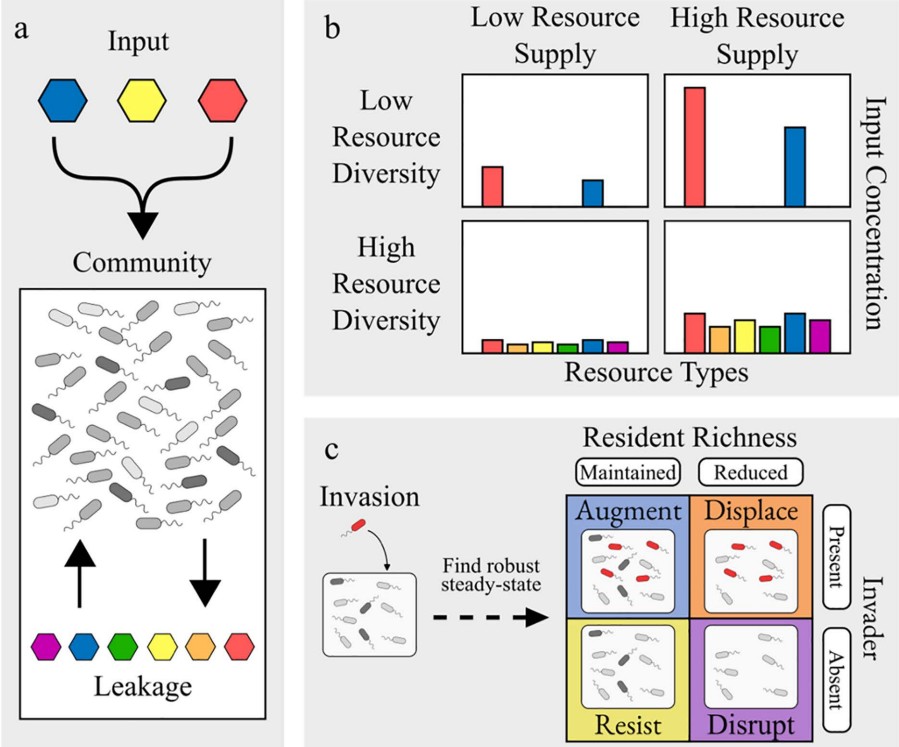

**Fig 1. Consumer-resource dynamics and invasion assay.** (a) Modeled resource flow, with input resources saturating the community of species which consume present resources and convert them to other resources via leakage, which can then be consumed by other species. (b) Closer look at resource input as modeled by supply and diversity. (c) Invasion assay and outcome assessment legend.

**Table 1. Dynamical variables and mechanistic parameters.**

| Parameter | Description | Value |
|---|---|---|
| $N_i$ | Population density of species $i$ (1/volume) | *Variable* |
| $R_\alpha$ | Concentration of resource $\alpha$ (energy/volume) | *Variable* |
| $K_\alpha$ | Input rate of resource $\alpha$ (energy concentration/time) | *0-18 (See Methods)* |
| $l$ | Leakage proportion (unitless) | 0.8 |
| $c_{i\alpha}$ | Uptake rate per unit of resource $\alpha$ by species $i$ (volume/time) | *0-10 (See Methods)* |
| $D_{\alpha\beta}$ | Proportion of resource $\beta$ converted to resource $\alpha$ (unitless) | *0-100% (See Methods)* |
| $m_i$ | Minimum energy uptake of species (energy) | 1 |
| $\gamma$ | Resource dissipation rate (1/time) | 1 |

This model is chosen based on the formulation in [38], in which two biologically relevant concepts are incorporated: (1) Energy considerations: each resource is assumed to carry a certain amount of energy, and a balance in energy expenditure is already in place as resources are converted to other resources and taken up by other consumers. (2) Metabolic trade-off: each consumer can utilize some of the resources, but the total uptake across all resources is fixed to capture the trade-off in resource utilization strategy. It should be noted that in this model, different resources are co-utilized and enough supply of any of the resources can support the growth of a species that can utilize that resource. The model, although simplified, is considered to be a reasonable abstraction of consumer-resource interactions among microbes and

has been used in several previous reports (e.g., [37,39,40]). Arbitrary units are used for resources and population densities in the model, and other parameters such as uptake rates are chosen accordingly in a range that would be consistent with typical microbial growth and uptake rates.

## Workflow and numerical analysis of consumer-resource model

The described microbial consumer-resource equations are encoded in MATLAB R2023a and numerically integrated using the ode45 nonstiff medium-order method with an absolute tolerance of $10^{-20}$. To maintain solver stability, communities are generated with at least one viable species and the resource supply is large enough to support a single ideal species. The assays described run on the Boston College Linux Cluster via Slurm scripting and using the MATLAB Parallel Computing toolbox. Raw data was imported to local machines to run analyses and create figures. Figures were generated using MATLAB and Inkscape.

## Parameter selection

For community generation, we set the following parameters (typical values listed in Table 2). We universally use $M = 18$ total resource types from which to choose supplied resource types and to which consumed resources may be converted. $E$ environmentally supplied resources are chosen at random from the $M$ total resource types, and $U$ concentration rate units are randomly allotted to the chosen supplied resource types.

Each column of the metabolic matrix is generated using the Dirichlet distribution with a parameter vector of size $M$ such that each element is $\frac{\sigma_D}{M}$, which ensures the columns sum to 1. Any element of $M \times M$ metabolic matrix less than the minimum conversion proportion $L_D$ is set to 0 and the columns are scaled to bring their sums to 1. Smaller metabolic spread $\sigma_D$ results in a more consolidated metabolism, while greater spread leads to greater resource diversity. For reference, metabolic spread values of 0.05, 1, and 5 result in an average of 1.13, 2.83, and 5.41 metabolic byproducts per consumed resource, respectively.

Once these environmental parameters are set, the community is colonized using a pool of $S_0$ species possessing random and unique resource preferences $c_{i\alpha}$. We sample $S_0$ as a random integer between 5 and 50 to generate a range of richness values (the justification for this sampling strategy is provided in JUSTIFICATION OF PARAMETER SELECTION). Each species has a preference budget $\mu_c$ of 10 uptake units to distribute among a maximum of 5 resource types, and a minimum zero-growth uptake of 1 energy unit. When a species consumes a single resource concentration unit, it produces $l$ units of metabolic byproducts and uses the remaining (1-$l$) units for growth. Invaders are generated in the same manner as initial

**Table 2. Community generation parameters.**

| Parameter | Description | Value |
|---|---|---|
| $M$ | Number of resource types in a simulated community | 18 |
| $U$ | Environmental resource supply (concentration/time) | 3-18 (See Methods) |
| $E$ | Environmental resource diversity (count) | 1-10 (See Methods) |
| $S_0$ | Initial species count (count) | 5-50 (See Methods) |
| $\mu_c$ | Total species uptake (volume/time) | 10 |
| $\sigma_D$ | Metabolic spread parameter | 0.05-5 |
| $L_{ext}$ | Consumer extinction threshold | 0.001 |
| $L_D$ | Minimum conversion proportion | 5% |
| $P_{inv}$ | Invader propagule proportion | 1% |
| $N$ | Number of simulated communities examined | 200,000-2,000,000 |

colonizers, with different resource preferences from any residents. After each propagation or dilution step, any species population less than the extinction threshold is set to 0 and subsequently removed from the system.

### Invasion assay

Once an invader is placed in a community, the community is propagated, and a numerical steady state is achieved. Next, the community undergoes a series of dilution-propagation steps. Lastly, iteratively perturbing population sizes and recomputing the steady state effectively filtered out most unstable species. To quantify colonization resistance in our invasion assay, we follow the same classifications as previous studies [22,34], illustrated in Fig 1c. We describe four categories based on the presence of the invader and resident richness in the final community. *Augmentation* describes when both the invader and every resident species are present in the final community; *displacement* describes when the invader but not every resident persists; *resistance* describes when the invader fails to colonize, and every resident species persists; and *disruption* describes when both the invader and at least one resident species go extinct.

We observe that with fixed resource input supply and diversity all tested communities reach a numerically stable steady state in which changes in species populations and resource concentrations are bounded. No non-diminishing oscillatory patterns were found after prolonged propagation steps, implying that stable consumer-resource steady states are achieved.

### Extinction threshold and disruption analysis

The model predicts that perturbations lead to loss of diversity in high-richness communities due to an increased number of populations near the extinction threshold. Upon invasion of these high-richness communities, disruption is the most common outcome, with the lowest-ranking species by abundance driven to extinction disproportionately more than expected by random chance (S1 Fig). This appears to be an inextricable element of the model, as varying the resource supply and resource diversity consistently leads to similar disruption phenomena. To investigate whether disruption outcomes can be reduced, we vary the extinction threshold under which species are considered extinct (S2 Fig). We find that extinction threshold does not significantly affect disruption outcomes, reinforcing that high-richness disruption is an inherent property of the consumer resource model.

### Justification of parameter selection

To determine the effect of initial species pool size on the resulting community, we compare richness and invasion outcome to $S_0$ (S3 Fig). We confirm that richness correlates with initial species count, validating our choice to use a range of pool sizes to generate communities of varying richness. Furthermore, resistance remains constant regardless of initial species count, implying colonization resistance is not significantly affected by this parameter. We test the effect of $M$, the number of possible resources in the community, on resistance outcomes, and find similar patterns as when $U$ and $E$ are varied (S4 Fig). Thus, we fix $M$ at 18 to reduce confounding variables. When introducing an invader to a community, its population size is determined relative to the total resident population. To determine the effect of propagule size $P_{inv}$ on invasion outcome, we tested different propagule sizes from $10^{-6}$ to $10^0$ of the resident community population. Our findings indicate little change in the proportion of resistance outcomes when the invader propagule size is greater than the extinction threshold of around $10^{-3}$ of the resident community population (S5 Fig). No additional significant propagule effects were found on the remaining invasion outcomes. As such, all other simulations use a constant invader propagule size of 1% of the resident community population.

### Invasion assay

To investigate invasion outcomes, we follow a two-step procedure. In the first step ('assembly' in Fig 2), a pool of species with randomly selected consumption and conversion profiles are grown together through several passages until a subset of

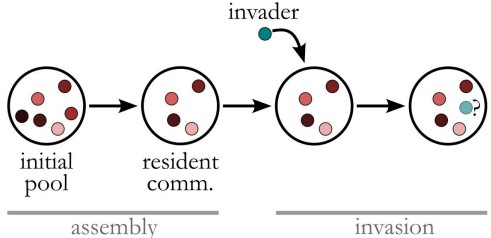

**Fig 2. Invasion into communities assembled from randomly interacting species is examined.** In the first stage, resident communities are assembled by allowing a pool of randomly interacting resources in the CR model to reach a steady state as the resident community. An invader is then introduced into the community and the community is simulated until it reaches a new steady state. The success/failure of the invader and a possible change in the richness of the resident members is used to categorize the invasion outcomes, as described in Fig 1.

these species reach stable coexistence. We then introduce an invader into these resident communities and examine follow the community to examine if the invader establishes in the community and if all the members of the resident community persist ('invasion' in Fig 2). We label the invasion outcomes as one of four possible outcomes (Augment, Displace, Disrupt, or Resist), according to the observed results in each particular instance of the simulation.

## Statistical analysis

The relative frequency of each invasion outcome is calculated by dividing the number of instances with that invasion outcome to the number of all simulated instances for that condition. To calculate the confidence intervals of the relative frequency of each invasion outcome, we used the `bootci` bootstrap routine in Matlab. For each data point, 95% confidence intervals are reported, using 1000 bootstrap samples.

## Results

### The richness-resistance relationship is not consistently monotonic

We begin by examining how resistance depends on the richness of resident communities. This dependency remains an outstanding question in invasion ecology due to inconsistent trends observed in different systems in previous investigations [20–23]. We consider a typical set of resource parameters (Eq (1) and Table 1) and analyze the relationship between species richness and the community's ability to resist invasion—termed the richness–resistance relationship. In this analysis, we quantify the relative frequency of observing different invasion outcomes (Augment, Displace, Disrupt, or Resist) when simulating many instances of invasion with parameters such as consumption rates and conversion rates are picked randomly from the corresponding distributions (Fig 2 and Table 1). We find this relationship to be non-monotonic (Fig 3a-3b), consistent with previous findings in a mediator-explicit model and a Lotka-Volterra model [22]. Furthermore, we observe a steady decline in augmentation outcomes, implying a reduction in unexploited niches with more present species. In its place, displacement steadily increases over the same richness domain as successful invaders more frequently push out residents in an overlapping niche. Lastly, disruption outcomes are insignificant at low richness but grow exponentially at high richness, showcasing the tendency to lose resident species in a packed community. To investigate the richness-resistance relationship further, we run the same experiment using varying levels of resource supply and diversity, measured by the total resource input and number of non-zero input resources, respectively.

### Resource diversity and supply shape the richness-resistance curve

Our results indicate that low resource supply environments have a strongly negative richness-resistance relationship due to high competition for few resources (Fig 4a). In contrast, high resource supply environments have a confounding,

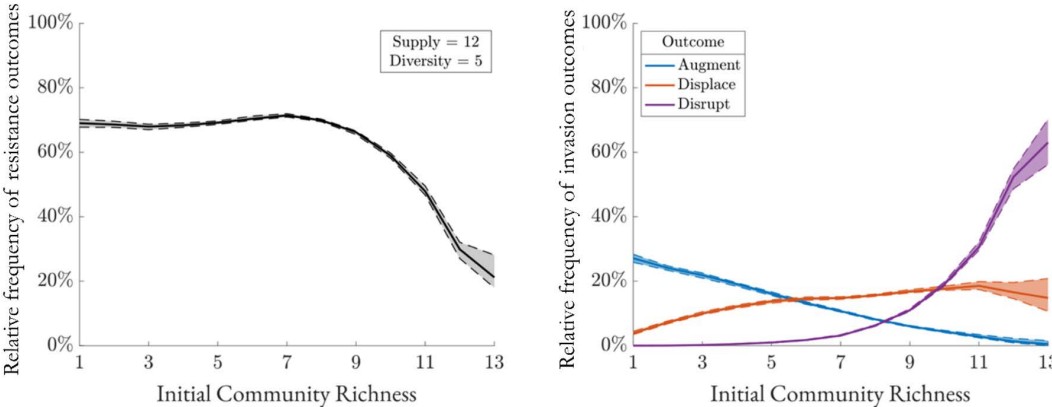

**Fig 3. The richness-resistance curve is not consistently monotonic.** (a) Communities are generated using specified resource supply and diversity parameters and categorized by richness. After invasion, proportion of resistance outcomes is plotted against richness. Dashed lines represent bootstrap 95%-confidence intervals (N = 200,000). (b) Proportions of non-resistance outcomes.

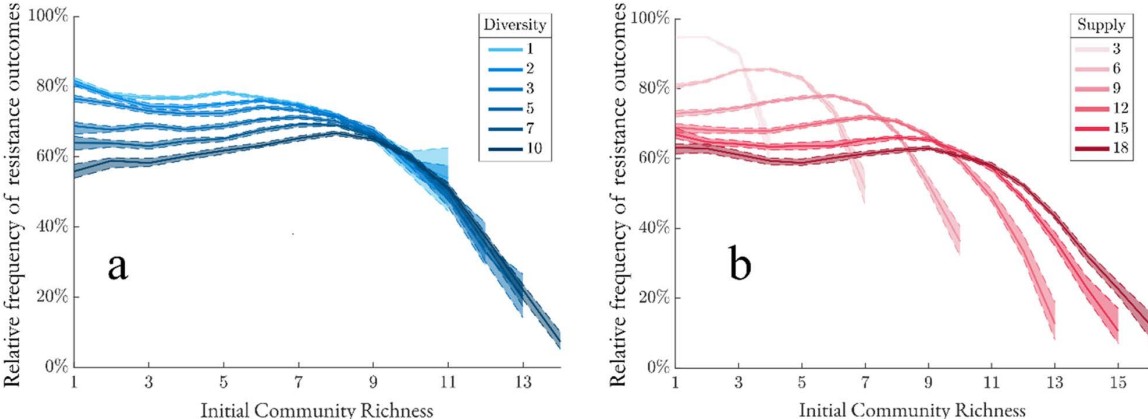

**Fig 4. Resource diversity and supply shape the richness-resistance curve.** (a) Resource diversity is fixed at 5 input resources and supply is varied (N = 1,000,000). (b) Resource supply is fixed at 12 concentration units and diversity is varied (N = 1,000,000). Dashed lines represent bootstrap 95%-confidence intervals. Larger confidence intervals at large values of initial community richness are because there are fewer communities emerging in our community assembly procedure at those richness values.

non-monotonic richness-resistance relationship at low species richness, but converge at higher richness to a similar negative trend as seen in resource-limited systems. In species-rich communities, population sizes near the extinction threshold tend to decrease, resulting in disruption overtaking resistance (Figs 3b and S1).

For all tested resource diversity values, the richness-resistance relationship is non-monotonic due to the abundant resource supply of 12 concentration units (Fig 4b). In low resource diversity environments, the richness-resistance relationship shows a negative correlation at low richness, a positive correlation at intermediate richness, and a negative correlation at high richness. While the general pattern persists at high resource diversity, the correlation in low-richness regimes reverses—from negative to positive—as resource diversity increases. Given the dynamic richness-resistance relationship under varying environmental inputs, we next analyzed the direct effect of resource supply and resource diversity on resistance.

### Increased resource supply leads to less resistant communities

Intuitively, increasing supply expands viable niche space, allowing more invaders to colonize and reducing resistance. Our findings agree with this hypothesis. Additional resource supply leads to less resistance outcomes regardless of resource diversity (Fig 5a). However, resistance in communities with greater resource diversity exhibits an increased sensitivity to resource supply.

### Increased resource diversity boosts resistance under limited resource supply, but reduces resistance otherwise

We then examined resistance against resource diversity (Fig 5b). We see that resistance generally decreases with resource diversity, and the decrease is more pronounced with greater resource supply. This further supports our hypothesis that increasing resource diversity or supply expands available niche space, leading to reduced resistance. Interestingly, in resource-limited environments, resource diversity positively correlates with resistance. Our data suggests that dispersing limited input resource units among many resource types restricts species from acquiring sufficient resources to meet the minimum energy uptake threshold.

### Cross-feeding weakens resistance especially when the resource input has low diversity

To evaluate the effect of the cross-feeding, we measure resistance in the four environments for low or high levels of byproduct leakage and metabolic spread (Fig 6a). Byproduct leakage is the proportion of consumption that is leaked as other resources ($l$ in Eq (1)), the distribution of which is determined by the metabolic matrix $D$ (Eq (1)). Notably, increased leakage decreases the amount of energy a consumer can use for growth, leading to lower overall population sizes in high leakage systems. Metabolic spread determines how byproduct leakage is distributed as other resources. Low-spread metabolism indicates that a consumed resource is converted into a few byproducts at high concentrations, whereas high-spread metabolism yields a broader range of byproducts in lower amounts.

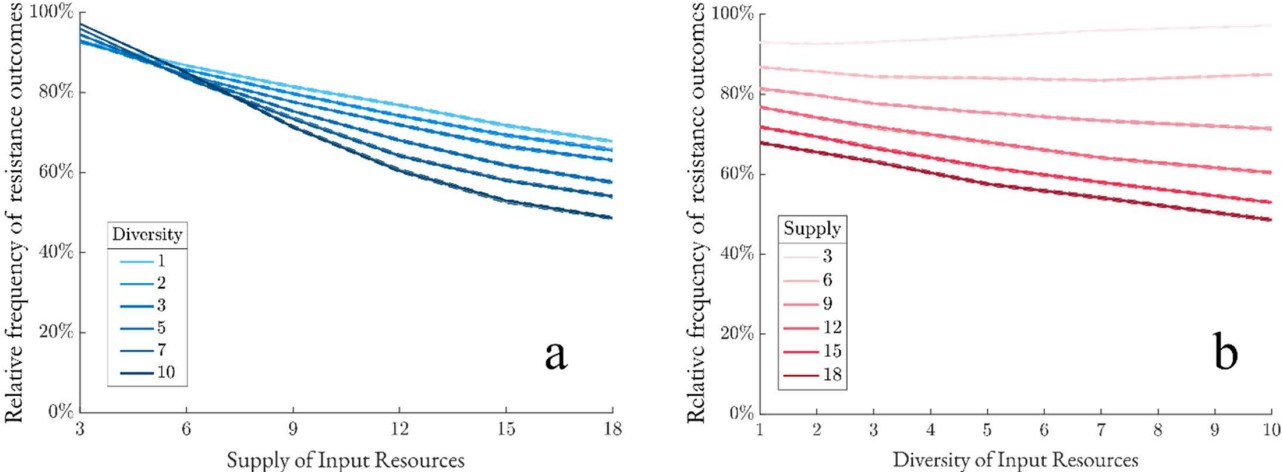

**Fig 5. Resistance exhibits mostly negative correlation with resource diversity and supply.** With resource supply and diversity parameters shown, communities are assembled and then exposed to invaders. (a) Communities are sorted by resource supply (darker lines indicating greater supply), then the relative frequency of resistance outcomes is plotted against resource diversity (N = 2,000,000). (b) Communities are sorted by resource diversity (darker lines indicating greater diversity), then the relative frequency of resistance outcomes is plotted against resource supply (N = 2,000,000). Dashed lines represent bootstrap 95%-confidence intervals.

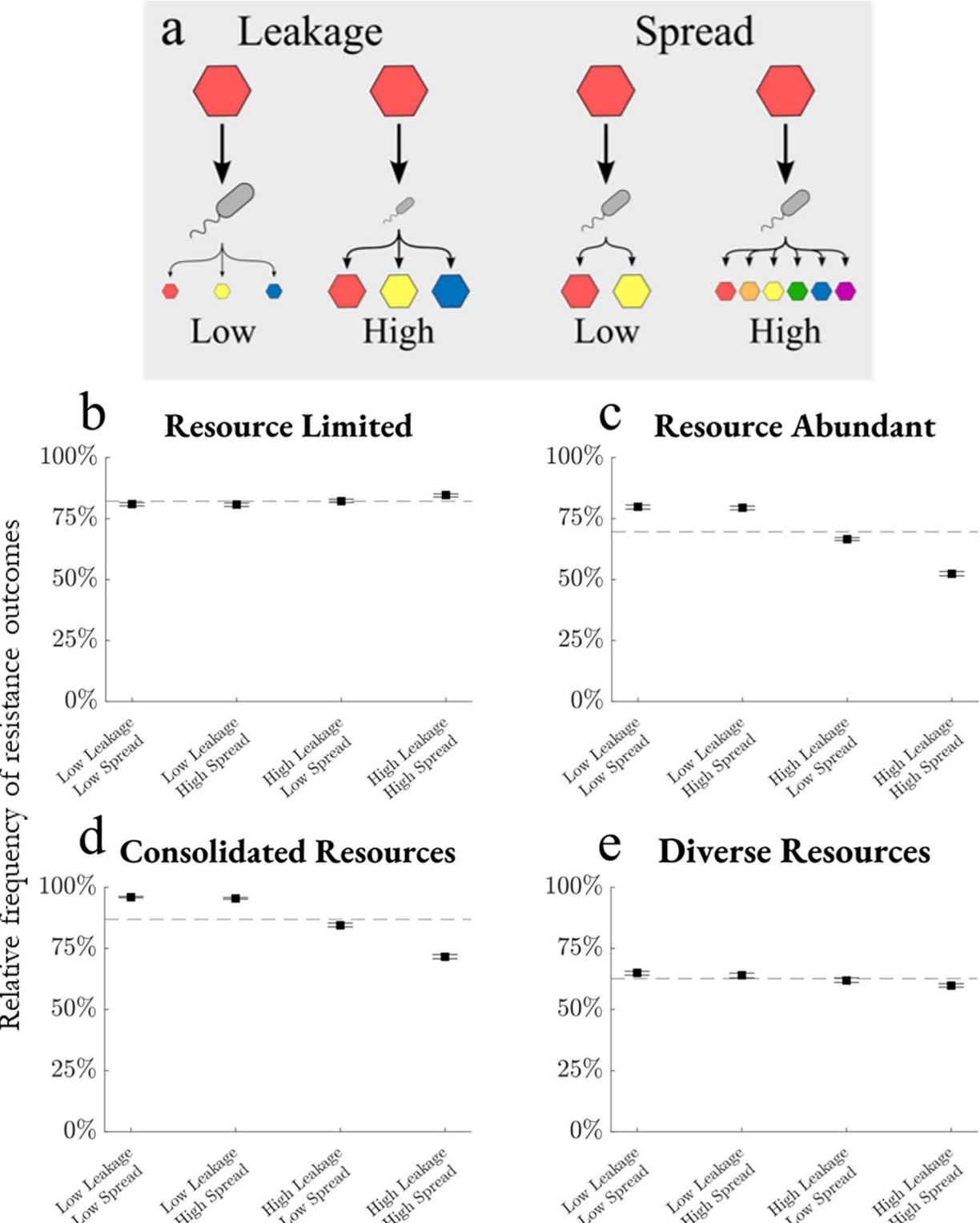

**Fig 6. Cross-feeding weakens resistance especially when input resource diversity is low.** (a) Low leakage results in more energy converted to growth and less resource output, while high leakage limits growth by converting more energy to other resources. Low spread metabolisms convert one resource to few others in large quantities, while high spread metabolisms result in several resource products at lower concentrations. (b-e) For each

environment, communities are generated with a low or high leakage ratio (I=0.01 or 0.8) and a low or high spread metabolism ($\sigma_D$=0.05 or 5). Communities are invaded and the outcomes are recorded, then total resistance proportion is calculated with 95%-confidence bootstrap intervals. Dashed lines are means over all cross-feeding parameters (N=200,000).

In environments not limited by resource supply, trends across byproduct leakage and metabolic spread are consistent (Fig 6c-6e). Low leakage increases resistance by allowing consumers to remove more resources from the environment and achieve nutrient blocking. Low spread also increases resistance by reducing the number of resource types present and constraining open niche space. Metabolic spread is only a factor when byproduct leakage is significant. Deviation from mean resistance is greatest when the resource diversity is low relative to resource supply, specifically in resource-abundant and consolidated resource environments.

Many of these observations are reversed in the resource-limited environment (Fig 6b). Here, high leakage and high spread each increase resistance, as opposed to the three other environments. Consistent with our previous results, increased resource diversity in resource-limited environments enhances resistance by making it more challenging for any consumer—whether resident or invader—to acquire sufficient resources for population maintenance. Because high byproduct leakage and metabolic spread each increase resource diversity, we see increased resistance in resource-limited environments and decreased resistance in each of the other cases. The effect of metabolic spread is still dependent on high byproduct leakage and deviation from the mean is small due to a low potential for creating new niches when resources are limited.

## Discussion

To investigate the effects of environmental resource supply and resource diversity on colonization resistance, we used a microbial consumer-resource model with co-utilized resources and explicit species resource preferences (Eq (1)). Previous studies have explored similar emergent functions using Lotka-Volterra or mediator-explicit models, but such models are not designed to account for the effect of resources in the environment. Furthermore, we incorporated cross-feeding dynamics using a leakage fraction parameter and metabolic matrix, allowing for resource-exchange interactions that shape community metabolism and remove nutrients from the environment. We expected variation in the degree of metabolic exchange across communities, so we tested the effect of byproduct leakage and metabolic spread on overall community resistance.

We discovered that a significant portion of our results could be predicted based on a few fundamental characteristics of the model: sensitivity at high richness, increased resistance due to nutrient blocking, and decreased resistance due to unutilized resources. We found a high sensitivity to perturbation at high richness in every tested environment, which resulted in disruption outcomes overtaking resistance. Moreover, the lowest rank-abundance species disproportionately account for extinction events in these conditions. These findings align with previous work using consumer-resource models which found neutral-like dynamics in communities with high species diversity, resulting in more frequent displacement and disruption events [41,42]. To further validate that the model dynamics are capable of replicating empirical studies, we analyze the resistance of the community relative to its survival fraction from the initial species pool. A recent study found that invasion success is approximately equal to the survival fraction of a community from its initial species pool [35], i.e., a community with 50% of the initial species present will admit an invader about 50% of the time. Using a constant initial species pool size, the consumer-resource model produces a similar trend (S6 Fig).

We note that disruption outcomes in this parameter space occur more frequently within the model than experimentally. Disruption can happen when communities with alternative steady states shift between states when a transient invader sufficiently perturbs the resident populations and/or resource concentrations [43]. An explanation for our observation is that experimental communities are generally more stable prior to perturbation than simulated ones in which extraneous species face less environmental filtering in the idealized model. Our results suggest a core community is assembled around the nutrients present, with such transient invaders surviving on the remaining scarce nutrients which are vulnerable to perturbation.

Nutrient blocking is a mechanism of colonization resistance in which resident species consume sufficient nutrients to block an invader from colonizing [17]. Environmental filtering is a similar process where an invader fails to colonize due to a lack of essential nutrients, driven by unsuitable abiotic conditions such as limited resource availability [44]. These two mechanisms drive increased resistance in communities with higher richness and environments with limited resource supply. Species-rich communities contain more metabolic strategies, resulting in increased resource consumption, nutrient blocking, and resistance. Likewise, limited resource environments provide fewer opportunities for an invader to realize a viable niche, resulting in stronger environmental filtering and increased resistance. In contrast, unutilized resources decrease resistance as more viable niches are available for an invader. High resource supply and resource diversity create open niches that invaders can exploit, weakening colonization resistance. Our cross-feeding experiments further support the relationship that resistance is strongly and negatively correlated with unutilized resources creating open niche space.

Cross-feeding dynamics offer an opportunity for the resident community to reshape environmental resource distributions. Increased byproduct leakage and metabolic spread effectively lead to higher resource diversity. Given a fixed resource supply, niche space is maximized when resource diversity is increased until resource concentrations are too diluted to support a population, a consequence of nutrient blocking and environmental filtering (Fig 7). In a resource-limited environment, increasing resource diversity via cross-feeding decreases niche space as fewer metabolic strategies remain viable, leading to slightly increased resistance with more cross-feeding. Similarly, in diverse resource environments, cross-feeding does little to increase niche space because the system is already saturated with supplied resource types, resulting in little change to resistance in response to changes in cross-feeding. However, cross-feeding in resource-abundant and consolidated resource environments increases niche space as sufficient resource supply is converted to many other resource types. This finding is experimentally supported by Dal Bello *et al.*, which found that high richness communities can be supported by a single carbon source, where cross-feeding dynamics facilitate a broad metabolic network incorporating numerous species [45]. In turn, resistance was most dynamic in the resource-abundant and consolidated resource environments, where higher leakage and spread led to increased resource diversity, niche space, and invasion success.

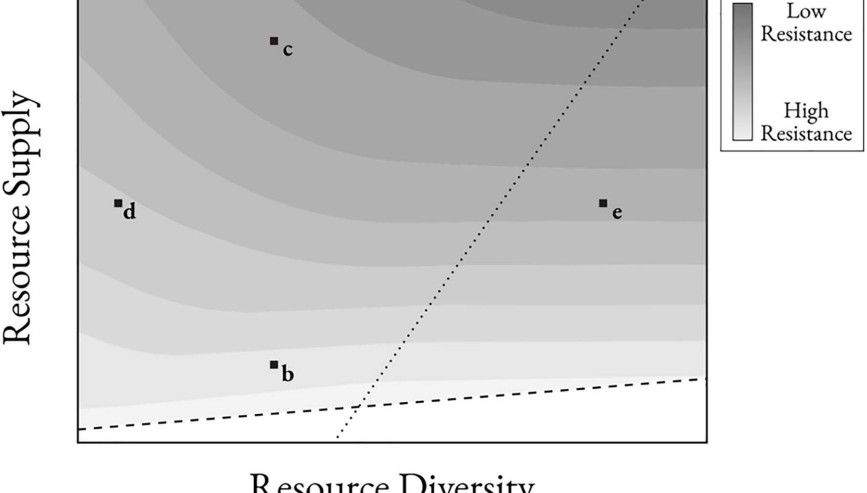

**Fig 7. Qualitative plot of resistance by resource diversity and supply.** Dotted line represents resource diversity threshold where environment is saturated with resource types, meaning additional resource diversity does not significantly impact resistance. Dashed line represents the maximum resource diversity without diluting concentrations beyond usable threshold, below which any species struggles to reach minimum uptake. Labels correlate with environments from Fig 6b-6e.

In this study, we classify different resistance regimes of resource input parameter space and show how the richness-resistance relationship is affected in these different regimes. Our work using the consumer-resource model reveals insights into how the environment shapes colonization resistance in microbial communities. Previous studies on invasion ecology often separate two determinants of invasion success: the available environmental niche that would be receptive to the invader and the biotic interactions, including the effect of resident community members, that can facilitate or inhibit the invader. Here we show that in a consumer-resource model, these two factors are tightly linked through the resources in the environment. This finding in particular highlights that resource supply—defined as the amount of resources supplied by the environment—and resource diversity—defined as the variety of supporting resources for consumers—as major determinants of invasion outcomes. Our findings offer an important insight and a new dimension for microbiome management strategies by controlling the amount and diversity of supplied resources into the environment when possible.

We recognize that the assumptions made for our model may limit its experimental applicability. One limitation is that the resources in our model are co-utilized and could, for example, represent alternative carbon sources. As such, the current model does not capture situations in which multiple resources are required simultaneously for growth (e.g., a carbon source and a nitrogen source). Additionally, parameterizing cross-feeding dynamics by a single, static matrix does not fully capture the diversity of possibilities such as independent resource requirements (e.g., carbon versus nitrogen sources), preferred utilization of certain resources [46], or metabolic plasticity [47]. We also assume a consistent environmental resource supply and allow communities to reach a steady state before perturbation and invasion occur, but in nature, microbial communities are not necessarily in equilibrium. Moreover, spatial structure and heterogeneity, the influence of the host, and evolutionary processes are not accounted for in our current investigation. These are important factors that need to be investigated in future studies. Despite these limitations, the consumer-resource model is an effective framework for predicting resource-driven niche space. Further investigation is needed to explore how spatiotemporal variation in resource supply and species populations impacts colonization resistance, both within consumer-resource models and in other theoretical frameworks.

## Supporting information

**S1 Fig. Disruption disproportionately affects communities of high richness and species of low abundance** .
(TIFF)

**S2 Fig. Selection of the extinction threshold.**
(TIFF)

**S3 Fig. Selection of the initial species count.**
(TIFF)

**S4 Fig. Resistance is not sensitive to the selection of the propagule size.**
(TIFF)

**S5 Fig. Community richness that shows maximal resistance depends on the number of resources in the model.**
(TIFF)

**S6 Fig. Survival fraction matches invasion success in our model, consistent with previous findings.**
(TIFF)

## Acknowledgments

We acknowledge computational resources and support offered by Boston College Research Services and the Linux Cluster staff.

## Author contributions

**Conceptualization:** Ethan S. Rappaport, Babak Momeni.

**Formal analysis:** Ethan S. Rappaport.

**Funding acquisition:** Renato Mirollo, Babak Momeni.

**Investigation:** Ethan S. Rappaport, Renato Mirollo, Babak Momeni.

**Methodology:** Ethan S. Rappaport, Babak Momeni.

**Software:** Ethan S. Rappaport.

**Supervision:** Renato Mirollo, Babak Momeni.

**Visualization:** Ethan S. Rappaport, Renato Mirollo, Babak Momeni.

**Writing – original draft:** Ethan S. Rappaport.

**Writing – review & editing:** Ethan S. Rappaport, Renato Mirollo, Babak Momeni.

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
