## [Decision Letter · Decision Letter 0]

22 Jul 2025

PCOMPBIOL-D-25-01039

Resource diversity and supply drive colonization resistance

PLOS Computational Biology

Dear Dr. Momeni,

Thank you for submitting your manuscript to PLOS Computational Biology. After careful consideration, we feel that it has merit but does not fully meet PLOS Computational Biology's publication criteria as it currently stands. Therefore, we invite you to submit a revised version of the manuscript that addresses the points raised during the review process.

Please submit your revised manuscript within 30 days Sep 21 2025 11:59PM. If you will need more time than this to complete your revisions, please reply to this message or contact the journal office at ploscompbiol@plos.org. Please include the following items when submitting your revised manuscript:

We look forward to receiving your revised manuscript.

Kind regards,

Jordan Douglas

Academic Editor

PLOS Computational Biology

Tobias Bollenbach

Section Editor

PLOS Computational Biology

**Additional Editor Comments:**

Thank you for submtiting this manuscript in its well-polished state. I have no doubt this study will be of interest to readers of PLOS Computational Biology. Please address the Reviewer's concerns below.

Regards,

Jordan Douglas

**Journal Requirements:**

4) We notice that your supplementary Figures are included in the manuscript file. Please remove them and upload them with the file type 'Supporting Information'. Please ensure that each Supporting Information file has a legend listed in the manuscript after the references list.

5) In the online submission form, you indicated that Data files used are available upon request.. All PLOS journals now require all data underlying the findings described in their manuscript to be freely available to other researchers, either

1. In a public repository

2. Within the manuscript itself

3. Uploaded as supplementary information.

**Reviewers' comments:**

Reviewer's Responses to Questions

**Comments to the Authors:**

Reviewer #1: # Reviewer remarks to the Author

This manuscript “Resource diversity and supply drive colonization resistance” tackles a fundamental question in microbial ecology: how environmental resource diversity and supply influence colonization resistance in microbial communities. Using a modified version of MacArthur’s consumer-resource modelling (captures the dynamics of species populations and resource concentrations over time) framework coupled with extensive computational invasion simulations, the authors provide insights into how resource factors modulate the relationship between species richness and colonization resistance, as well as the role of cross-feeding dynamics.

This study explores how environmental factors shape the relationship between species richness and colonization resistance. It finds that this relationship is non-monotonic and influenced by both resource supply and diversity. Resistance generally decreases with higher resource supply and often with greater resource diversity—except under resource-limited conditions, where increased diversity surprisingly enhances resistance. Additionally, cross-feeding—which increases resource diversity via metabolic byproducts—can weaken resistance, though this effect vanishes when resources are scarce. Overall, the findings highlight how both external resource inputs and internal metabolic interactions shape a community’s ability to resist invasion. This work advances prior studies focused mainly on species interactions by incorporating explicit resource dynamics, offering valuable perspectives for both natural and engineered microbiomes.

As an experimental biologist, I approach this work from a practical and empirical perspective. While I recognize the value of the modeling framework, my review focuses on the biological relevance of the assumptions and findings, and how well they align with observed microbial behavior and potential for experimental validation and draw parallels within and across microbiomes.

Key contributions

• Non-monotonic richness-resistance relationship: This study reveals that colonization resistance does not simply increase with species richness but follows a complex, resource-dependent pattern, helping to reconcile conflicting empirical observations.

• Cross-feeding effects: Cross-feeding increases resource diversity and can weaken resistance, but this effect disappears under resource limitation, highlighting nuanced ecological interactions relevant for microbiome manipulation.

• Clear communication: The hypotheses, methods, and results are well-articulated, making the study accessible to a broad scientific audience.

Strengths

• Advanced modeling approach: The integration of consumer-resource dynamics with cross-feeding and invasion assays offers a systematic understanding beyond classical ecological models.

• Robust computational design: The authors simulate millions of community replicates with bootstrap confidence intervals, ensuring strong statistical support.

• Relevance: The work addresses the “invasion paradox” by demonstrating how resource supply and diversity condition the effect of richness on invasion resistance.

• Reproducibility: Code availability via GitHub enhances transparency and facilitates future research.

Limitations and suggestions for improvement

Model realism and assumptions

• The model abstracts away spatial heterogeneity, host influences, and evolutionary processes. These simplifications should be explicitly acknowledged, with discussion on how they may affect the generalizability of conclusions.

• The parameterization of resource types, supply rates, and cross-feeding interactions requires stronger empirical grounding or comparison to real microbial systems.

• Empirical validation: This study is purely computational. Incorporating comparisons with experimental or observational data or at least discussing how the predictions align with existing empirical findings would significantly strengthen the manuscript.

• Parameter sensitivity: Beyond leakage and cross-feeding spread, additional parameters (e.g., resource uptake rates, species pool composition, extinction thresholds) deserve systematic sensitivity analyses to confirm robustness.

Statistics

Unfortunately, the provided information does not sufficiently evaluate whether the statistics performed in the paper are sound and well-written. None of the results appear to directly quote or discuss the statistical methods section of the paper.

Authors should include:

• Appropriateness of the statistical tests used for the data types and experimental designs

• Adequate description and justification of the statistical models and assumptions

• Proper handling of replicates, controls, and data normalisation procedures

• Reporting of effect sizes, confidence intervals, and statistical power

• Data processing and analysis workflows and significance testing, where appropriate.

Broader context

The findings contribute meaningfully to ecological theory by elucidating how environmental resource factors mediate colonization resistance and community invasibility. Expanding the discussion to connect with broader invasion ecology literature, empirical work, and potential applications, such as microbiome management strategies, would enhance the manuscript’s impact.

Overall manuscript presents a exciting theoretical framework with interesting insights into microbial colonization resistance. Addressing the above points—particularly empirical validation, parallels between this and other studies, expanded sensitivity analyses, and improved clarity—will elevate the work’s rigour and relevance, making it suitable for publication.

Reviewer #2: Summary:

This article investigates how environmental resource diversity and supply influence colonization resistance in microbial communities, utilizing a consumer-resource model and computational invasion simulations. The authors explore the richness-resistance relationship, overall colonization resistance, and cross-feeding dynamics. Key findings include a non-monotonic relationship between species richness and resistance, which is influenced by environmental characteristics. They found that colonization resistance is negatively correlated with both resource supply and diversity, except when resource supply is limited. Additionally, cross-feeding generally weakens colonization resistance by increasing resource diversity, but this trend diminishes under limited resource supply. The study highlights the importance of nutrient blocking and environmental filtering as mechanisms driving increased resistance in resource-limited or species-rich environments.

Major Critiques:

Methods

Numerical analysis: The authors state that the consumer-resource model was encoded in MATLAB and solved using ode45, and that all results are based on these numerical integrations. However, it remains unclear how the simulations map onto each key result (e.g., Figs 3–5). For example, how exactly were resistance proportions computed as a function of supply or diversity? Were species initialized with a fixed or randomized richness for each trial? Was the invader always introduced at the same abundance and time point? A summary table or flowchart outlining which parameters were varied in each experiment, how outcomes were classified, and how these relate to the model equations would greatly improve clarity and reproducibility.

Microbial consumer-resource model / parameter selection: Authors did not represent what the 18 resources types represent. Are these different carbon source, or a mix of carbon, nitrogen, and other nutrients? This abstraction limits the biological interpretation of resource diversity and cross-feeding. The concept of “environmental diversity” is central to the manuscript but left undefined in biological terms. The authors randomly select E resource types from a pool of 18, but the nature of those resources remains unclear. Without examples, it is difficult to interpret whether high diversity refers to nutrient-rich gut environments, diverse carbon substrates. A qualitative mapping or explanatory figure is strongly recommended.

Model assumes static resource preferences and a fixed metabolic matrix for all species, which may not capture the metabolic plasticity and adaptive behaviors observed in real microbial communities.

The model presumes a constant environmental resource supply and equilibrium before invasion, whereas natural communities often experience fluctuating conditions and may not reach steady state.

The model does not account for other important ecological interactions such as antimicrobial production, spatial structure, or host factors, which can significantly influence colonization resistance.

Minor Critiques:

Although the manuscript introduces two differential equations at the heart of the consumer-resource model, these equations are never referenced in the text, nor are they numbered. This weakens the link between the mathematical model and the ecological interpretations presented. I strongly recommend the authors number the equations and refer back to them consistently in the Methods, Results, and Discussion when interpreting effects of parameters such as resource supply, cross-feeding, and diversity.

Reviewer #3: Understanding the relationship between community diversity / richness and ability to resist invasion is a long-standing challenge in ecology and microbiology. This paper uses a consumer-resource model with cross-feeding to explore how environmental factors—specifically resource diversity and supply—affect colonization resistance in microbial communities. The use of a resource explicit model allows the authors investigate the richness-resistance relationship and how it depends on resource supply and diversity. The authors argue that there is frequently a non-monotonic richness-resistance relationship that is shaped by environmental conditions. Overall, I found the paper to be interesting, although as discussed below I wouldn’t characterize the observed curves as being non-monotonic, as any initial increase in resistance with richness is very modest.

Major points:

Figure 2a: What is “resistance outcomes”? From the text, my guess is that it is the fraction of invasions that are successful? This is very important for readers to understand, and the phrase resistance outcomes is very unclear to me. In any case, I would not describe this curve as being non-monotonic, as any increase is very slight. Instead, I would say that it is flat then decreasing.

The case of disruptions is very interesting to me. Is this caused by alternative stable states in the original community? It reminds me of the case of “transient invaders” that we characterized experimentally (Amor et al, Science Advances (2020)). In any case, the simplest community dynamics would not be expected to lead to disruptions, so understanding the dynamical origin of this is interesting and important. In the cases of disruption, do you see propagule effects (eg outcomes that depend on the initial population size of the invader)?

We recently found that the probability of a successful invasion (in the absence of priority effects / alternative stable states) was approximately equal to the survival fraction during assembly of the original community (Hu et al, Nature Eco & Evo (2025)). Is this something that is true in your model? (In your case you start with some number S_0 species then achieve some stable richness, so you have the data to calculate this survival fraction)

The model makes many particular decisions (see minor points below), and it was difficult for me to always know when each decision is relevant for any given outcome. To some degree this is inevitable, but the problem is compounded the more additions there are to the model. For example, the minimum resource uptake for survival is presumably required for some of the results at low resource supply (which the authors acknowledge), but there were many such questions in my mind (dilution needed, etc).

Minor points:

I didn’t understand how resource preferences were implemented in the model. Is there co-utilization of resources based simply on the c_i,alpha values? Or do species focus on resources based on a resource preference order?

Gamma is called the environmental diffusion rate, which strikes me as confusing. Isn’t it just the resource degradation rate?

“We universally use 18 total resource types from which to choose supplied resource types and to which consumed resources may be converted. E environmentally supplied resources are chosen at random from the M total resource types, and U concentration rate units are randomly allotted to the chosen supplied resource types.” I don’t understand the difference between 18 and M here.

I don’t see how the minimum of one unit of resource uptake for zero growth is encoded in the model. It would be nice if the authors would comment on the importance of this assumption, as it is not standard in these resource models (although is indeed frequently observed in the starvation regime).

Given that the authors have included a resource degradation rate (and species mortality), is it necessary to also assume a dilution step?

**Have the authors made all data and (if applicable) computational code underlying the findings in their manuscript fully available?**

Reviewer #1: Yes

Reviewer #2: Yes

Reviewer #3: Yes

PLOS authors have the option to publish the peer review history of their article (what does this mean? ). If published, this will include your full peer review and any attached files.

**Do you want your identity to be public for this peer review?** For information about this choice, including consent withdrawal, please see our Privacy Policy .

Reviewer #1: **Yes: ** Samir Giri

Reviewer #2: **Yes: ** Rajib Saha

Reviewer #3: **Yes: ** Jeff Gore

**Figure resubmission:**
---

## [Decision Letter · Decision Letter 1]

22 Oct 2025

Dear Dr. Momeni,

We are pleased to inform you that your manuscript 'Resource diversity and supply drive colonization resistance' has been provisionally accepted for publication in PLOS Computational Biology.

Best regards,

Jordan Douglas

Academic Editor

PLOS Computational Biology

Tobias Bollenbach

Section Editor

PLOS Computational Biology

Reviewer's Responses to Questions

**Comments to the Authors:**

Reviewer #1: This revised version of the manuscript has significantly improved in terms of clarity. The authors have satisfactorily

addressed my previous comments, and I have no further comments.

**Have the authors made all data and (if applicable) computational code underlying the findings in their manuscript fully available?**

Reviewer #1: Yes

PLOS authors have the option to publish the peer review history of their article (what does this mean? ). If published, this will include your full peer review and any attached files.

**Do you want your identity to be public for this peer review?** For information about this choice, including consent withdrawal, please see our Privacy Policy .

Reviewer #1: **Yes: ** Samir Giri

---

## [Editor Report · Acceptance letter]

PCOMPBIOL-D-25-01039R1

Resource diversity and supply drive colonization resistance

Dear Dr Momeni,

I am pleased to inform you that your manuscript has been formally accepted for publication in PLOS Computational Biology. Your manuscript is now with our production department and you will be notified of the publication date in due course.

With kind regards,

Zsofia Freund
